# High-resolution distribution maps of single-season rice in China from 2017 to 2022

Ruoque Shen[1,2], Baihong Pan[3], Qiongyan Peng[1,2], Jie Dong[4], Xuebing Chen[1,2], Xi Zhang[1,2], Tao Ye[5], Jianxi Huang[6], and Wenping Yuan[1,2]

[1]International Research Center of Big Data for Sustainable Development Goals, Beijing 100094, China

[2]School of Atmospheric Sciences, Southern Marine Science and Engineering Guangdong Laboratory (Zhuhai), Sun Yat-sen University, Zhuhai 519082, Guangdong, China

[3]Department of Microbiology and Plant Biology, University of Oklahoma, Norman, OK, 73019, USA.

[4]College of Geomatics & Municipal Engineering, Zhejiang University of Water Resources and Electric Power, Hangzhou 310018, Zhejiang, China

[5]Key Laboratory of Environmental Change and Natural Disaster, Ministry of Education, Beijing Normal University, Beijing 100875, China

[6]College of Land Science and Technology, China Agricultural University, Beijing 100083, China

*Correspondence to*: Wenping Yuan (yuanwp3@mail.sysu.edu.cn)

**Abstract**. Paddy rice is the second-largest grain crop in China and plays an important role in ensuring global food security. However, there is no high-resolution map of rice covering all of China. This study developed a new rice mapping method by combining optical and synthetic aperture radar (SAR) images in cloudy areas based on the time-weighted dynamic time warping (TWDTW) method and produced distribution maps of single-season rice in 21 provincial administrative regions of China from 2017 to 2022 at 10 or 20-m resolution. The accuracy was examined by using 108195 survey samples and county-level statistical data. On average, the user's, producer's, and overall accuracy over all investigated provincial administrative regions were 73.08 %, 82.81 %, and 85.23 %, respectively. Compared with the statistical data from 2017 to 2019, the distribution map explained 83 % of the spatial variation of county-level planting areas on average. The distribution maps can be obtained at https://doi.org/10.57760/sciencedb.06963 (Shen et al., 2023).

**Keywords**: rice; single-season rice; time-weighted dynamic time warping; Sentinel

## 1. Introduction

As the fourth-largest grain crop in the world, rice contributed 8 % to world food production in 2019 (FAO, 2021). Rice is a staple food for more than half of the world's population and plays an important role in ensuring global food security (Elert, 2014; Kuenzer and Knauer, 2013). The flooding of rice paddy fields constitutes a major source of methane emissions (IPCC, 2022; Mohammadi et al., 2020). Therefore, quickly and accurately identifying the planting location of rice over a large area is very important.

Most commonly, large-scale crop mapping takes advantage of satellite data (Dong et al., 2020; Huang et al., 2022; Xiao et al., 2006, 2005). Popular crop-mapping methods are various machine learning methods, such as random forest (Boryan et al., 2011; Fiorillo et al., 2020; You et al., 2021), support vector machine (Zheng et al., 2015), and deep learning (Thorp and Drajat, 2021; Zhao et al., 2019; Zhong et al., 2019). Machine learning methods provide several advantages in crop mapping

but require training samples (Belgiu and Csillik, 2018), commonly in the order of hundreds or even thousands to obtain a satisfactory accuracy (Millard and Richardson, 2015; Valero et al., 2016). For example, the Cropland Data Layer (CDL) products produced by the U.S. Department of Agriculture (USDA) use tens of thousands of training samples to map the crops of a single state (Boryan et al., 2011). Therefore, such large-scale investigations are very time-consuming and labor-intensive.

Another crop mapping approach is based on the detection of specific phenological signals. Xiao et al. (2005, 2006)

produced a 500-m resolution rice map of Southern China, Southeast Asia, and South Asia using MODIS (Moderate Resolution Imaging Spectroradiometer) data by comparing the Normalized Difference Vegetation Index (NDVI) and the Enhanced Vegetation Index (EVI) with the Land Surface Water Index (LSWI). In addition, Dong et al. (2016) also used the flood-detection method, producing a rice map with 30-m spatial resolution in Northeast Asia based on Landsat-8 data. Because of the short flooding period, the influence of clouds and rain in a few images will lead to missing the flooding signal and decreased

accuracy, placing high requirements for image quality and time resolution (Dong et al., 2016).

Additional crop mapping approaches are the dynamic time warping (DTW) and the time-weighted dynamic time warping (TWDTW) methods, which do not consider the crop characteristics within a certain time period but compare the signals over an extended period (Belgiu and Csillik, 2018; Qiu et al., 2017; Skakun et al., 2017; Zheng et al., 2022b). Guan et al. (2016) mapped rice in Vietnam using the DTW method based on MODIS NDVI data, reaching an $R^2$ of 0.809. The TWDTW method,

which is an improvement of the DTW method, adds a time weight to the calculation to characterize the temporal difference, improving identification accuracy (Maus et al., 2016). The TWDTW method has been used in several studies to produce high-resolution crop maps of many kinds of crops, including winter wheat, sugar cane, and maize (Dong et al., 2020; Huang et al., 2022; Zheng et al., 2022a; Shen et al., 2022). A previous study also used the TWDTW method to produce a map of double-season paddy rice in China by using the vertical-horizontal (VH) band signal from the Sentinel-1 satellite (Pan et al., 2021).

Because of the flooding during rice planting, the traditional rice mapping methods use water indexes derived from optical images, such as LSWI (Xiao et al., 2002, 2005). However, optical images are greatly impacted by clouds, heavily limiting their availability in cloudy regions (Li and Chen, 2020; Sudmanns et al., 2020; Zhou et al., 2019). An alternative is the use of synthetic aperture radar (SAR) images. Compared with the optical signal, the SAR signal can penetrate through clouds, completely avoiding their influence (Oguro et al., 2001; Phan et al., 2018). Several studies have demonstrated the capability

of SAR in rice identification and obtained good mapping results at the regional scale (Nguyen et al., 2016; Han et al., 2021;

Pan et al., 2021). However, compared with optical data, SAR data also has more significant salt-and-pepper noises, which may affect the accuracy of the distribution map (Oliver and Quegan, 2004).

China is the world's largest rice producer, producing 209.61 million tons of rice in 2019 (*China Statistical Yearbook*, 2020). Except for a few provinces in Southeastern China, most of the rice-planting provinces are dominated by single-season rice. Although there are many previous studies on mapping rice in China, a high-resolution single-season rice map is still not available for the entire country. This study attempts to fill this gap and aims to: (1) develop a new phenology-based method for rice mapping; (2) produce high-resolution distribution maps of single-season rice in China from 2017 to 2022; (3) evaluate the accuracy of the identified areas using county-level statistical data and survey samples.

## 2.  Data and methods

### 2.1   Study area

This study was conducted in 21 provincial administrative regions in mainland China, where the total planting area of single-season rice was 19.92 million hectares, accounting for approximately 99.01 % of the total planting area of single-season rice in mainland China according to the statistical data in 2018 (https://data.stats.gov.cn). The total production of the single-season rice in the study area was 150.46 million tons, accounting for approximately 98.91 % of the total production in mainland China in 2018. As single-season rice is wildly planted in China, this study further divided the study area into four subregions (Fig. 1). Subregion I is the northern rice planting area, including Heilongjiang, Jilin, Liaoning, Inner Mongolia, and Ningxia. Because of temperature limitations, only single-season rice is planted in this subregion, with the transplanting period generally between late May and mid-June. Subregion II is the middle southern single-season rice planting area, including provinces that only or mainly plant single-season rice (Jiangsu, Anhui, Hubei, Henan, Shandong, Shaanxi, and Shanghai) and provinces where single-season and double-season rice are both planted (Hunan and Jiangxi). The single-season rice in this subregion is generally transplanted between mid-late May and late June. Subregion III is the southeastern coastal single-season rice planting area, including Zhejiang, Fujian, and Guangxi. Here, single-season rice may be transplanted later than in Subregion II, generally between mid-late May and early July. Subregion IV is the southwestern rice planting area, including Sichuan, Yunnan, Guizhou, and Chongqing. Single-season rice in this subregion is transplanted much earlier than in other subregions, generally between late April and mid-May.

## 2.2    Data

### 2.2.1    Satellite data and landcover data

The satellite data used in this study were from the Sentinel series launched by the European Space Agency (ESA). The SAR data were obtained from the Ground Range Detected (GRD, Level-1) product of Sentinel-1A, and the optical data were obtained from the Level-1C product of Sentinel-2. The SAR data used in this study were the VH band (dual-band cross-polarization, vertical transmit/horizontal receive) at a spatial resolution of 10-m, and was composited into a 12-day temporal resolution by median. Optical data included ten bands (blue, green, red, and near infrared (NIR) at a 10-m spatial resolution, and red edge1 (RE1), red edge2 (RE2), red edge3 (RE3), red edge4 (RE4), shortwave infrared1 (SWIR1), and shortwave infrared2 (SWIR2) at a 20-m spatial resolution). Additionally, two indexes, NDVI and LSWI at a 10-m spatial resolution, were calculated with the following equations:

$$NDVI = \frac{\rho_{NIR} - \rho_{red}}{\rho_{NIR} + \rho_{red}} \tag{1}$$

$$LSWI = \frac{\rho_{NIR} - \rho_{SWIR1}}{\rho_{NIR} + \rho_{SWIR1}} \tag{2}$$

where $\rho_{NIR}$, $\rho_{red}$, $\rho_{SWIR1}$ are the reflectances of the NIR, red, and SWIR1 bands of Sentinel-2, respectively.

The Sentinel-2 Cloud Probability (S2C) product (https://developers.google.com/earth-engine/datasets/catalog/COPERNICUS_S2_CLOUD_PROBABILITY) was used to eliminate the influence of clouds. The product provides a cloud probability from 0 to 100 at a 10-m resolution, which has a higher resolution than the original QA60 band of the Sentinel-2 dataset and is more flexible and accurate. In this study, the threshold of cloud probability was set to 50; pixels with a higher probability were regarded as clouds and removed. Considering the length of the transplanting period and the number of cloud-free images of each subregion, the optical data were composited to 12-day (Subregion I) or 6-day (Subregion II, III, and IV) temporal resolution by median. Figure 2 shows the percentage of good optical observations during the study period of each pixel in the study area. A linear interpolation was applied to fill the gaps in the time series. To further eliminate the noise in the time series of Sentinel-1 and Sentinel-2 images, a Savitzky-Golay (SG) filter with the order set to two and the window size set to five was applied to smooth the time series. (Chen et al., 2004). All the preprocessing was completed on the Google Earth Engine (GEE) platform (Gorelick et al., 2017). Besides, this study used the Finer Resolution Observation and Monitoring of Global Land Cover (FROM-GLC) product as a mask to exclude non-cultivated areas (Gong et al., 2019).

### 2.2.2    Field data and agricultural statistical data

The field data were obtained through several field surveys we conducted across China during 2017–2021, including 37036

samples of single-season rice and 71159 samples of other crops (double-season rice, maize, soybean, peanuts, etc.), forests, built-up areas, water bodies, etc. An unmanned aerial vehicle (UAV; eBee, senseFly Ltd., Switzerland) was used in some of our surveys to take very-high-resolution images covering on average 0.1 km$^2$. The images were visually interpreted to obtain sample points at a spatial resolution of 20 m. The province-level agricultural statistical data are published in the statistical yearbook of each province, and the county-level statistical data are published sporadically in the statistical yearbook of each province or city. Since the release of data usually lagged by two years or more, and the single-season rice planting areas were not published in many counties, this study only collected a total of 2748 county-level statistical single-season rice planting area data from 2017 to 2019 (Table 1). No available county-level statistical data were collected for Heilongjiang and Inner Mongolia due to the discrepancies between the administrative division and statistical caliber.

## 2.3    Method

Figure 3 shows the flow of the single-season rice mapping method proposed in this study, including four steps: (1) preprocess of the Sentinel data; (2) calculate the distances of SAR and optical bands separately using the TWDTW method with translation and stretching; (3) combine the distances of the two bands using a weighted sum; (4) generate the distribution map using a threshold determined by the provincial-level statistics.

### 2.3.1    Time-weighted dynamic time warping method

This study generated the single-season rice distribution map by comparing the dissimilarity of the time series of each pixel with the standard time series of rice. The TWDTW method was used to calculate the dissimilarity (Petitjean et al., 2012; Dong et al., 2020). In this method, the unknown time series is non-linearly stretched or compressed to align with the standard single-season rice time series, and an accumulated distance is calculated by cumulating the distance of the alignment path. The accumulated distance of all possible alignments is calculated, and the minimum accumulated distance is used to represent the dissimilarity of two time series. Considering the phenophases of crops, a penalty called time weight is added to the calculation (Maus et al., 2016). When the time series is stretched or compressed, the difference caused by the dislocation of time axes is calculated, and a function (e.g., logistic function) is used to convert the time difference into a time weight. As a result, the TWDTW measures the difference between two time series by considering both shape and phenological information. Parameters of the TWDTW used in this study were suggested by Belgiu and Csillik (2018), by using a logistic function with $\alpha$ and $\beta$ set to 0.1 and 50, respectively. Finally, the single-season rice was identified through a threshold of dissimilarity determined by the province-level statistical area. The total area of pixels with dissimilarity lower than the threshold was equal to the statistical area.

### 2.3.2 Optical bands selection

The common method of rice establishment is transplanting. Rice seeds are first planted in a small field or a nursery, and then transplanted to the main field after the rice seedlings reach the three-leaf stage. The transplanting method can be divided into machine transplanting, manual transplanting, and seedling-throwing. All the transplanting methods require the field to be flooded, which is the main feature that distinguishes rice from other crops. Figure 4 shows the time series of all 10 optical bands or indexes of four main crops in Jilin Province in 2019. It can be seen that the time series of rice of three moisture-related bands or indexes, including SWIR1, SWIR2, and LSWI are significantly different from those of other crops during the transplanting period (DOY 133–181). LSWI is designed to characterize land surface moisture, and its value positively correlates with land surface moisture, showing a high value during the transplanting period. In contrast, the two SWIR bands show the opposite, first decreasing and then increasing during the transplanting period, following a "V" shape. As the aim of this study was to map the distribution of single-season rice, the time series did not necessarily need to be able to characterize a certain land surface parameter like LSWI. The priority was whether a band or index could distinguish rice from other crops. As LSWI is calculated as the normalized difference of the NIR and SWIR1, and the NIR of rice also decreases slightly during the transplanting period, offsetting the LSWI increase caused by the decrease of SWIR1, erasing some differences and information. The change of SWIR2 was less pronounced than that of SWIR1, so SWIR1 was selected in this study to calculate the dissimilarity.

The standard time series were generated using survey samples. Fifty single-season rice survey samples were randomly selected from all single-season rice points in each province. The SWIR1 time series of these samples were extracted, aligned according to the time when the minimum value appears, and averaged to obtain the standard time series of each province. The standard time series of 21 provincial administrative regions in four subregions all showed a "V" shape (Fig. 5). The time period of the standard time series was limited to the transplanting period, and the length of the standard time series were five in Subregion I and seven in other subregions. Because the method is transferable between years, the standard time series retrieved from one year was used in all six years.

### 2.3.3 TWDTW with translation and stretching

Although the transplanting period of rice is short, farmers may transplant single-season rice over a longer period. Therefore, the "V" shape may appear earlier or later. In this study, the standard time series was translated and stretched along the time axis to match any possible period in which transplant might have occurred. To reduce the computational effort as well as to prevent overstretching, the standard time series was allowed to stretch for at most one time. Taking Subregion I as an example, where rice is generally transplanted from late May to mid-June, the study period was set to day of year (DOY)

121–193, with a total of seven observations to take decreasing and increasing phases of the "V" curve into consideration. The length of the standard time series was five, and time series within the study period with a length of five or six were selected to calculate the distance with the standard time series using the TWDTW method (Fig. 6). The minimum of all distance was selected to represent the dissimilarity of the pixel. Study periods of Subregion II, III, and IV were set to DOY 121–199, 121–211, and 97–163, respectively.

### 2.3.4    Combine SAR images in Southern China

Compared with northern China, southern China is more heavily affected by clouds and rain, resulting in a worse quality of optical observation (Fig. 2). This study introduced SAR observations as an auxiliary in Subregion II, III, and IV, as it can pass through clouds. Specifically, the VH band was used because studies have shown that VH polarization is more sensitive than VV in detecting field flooding (Nguyen et al., 2016; Wakabayashi et al., 2019). The VH time series of rice in the transplanting period also shows a "V" shape (Fig. 7). Although the shape of the rice curve differs from that of the other crops, their values partly overlap. As a coherent radar system, SAR images will inevitably carry salt-and-pepper noises (Veloso et al., 2017). Therefore, VH was only used when the quality of optical observation was extremely poor.

First, the dissimilarity of an unknown VH time series and the standard VH time series were calculated at each pixel using the TWDTW method. The standard VH time series was generated using the same procedure as for SWIR1 (section 2.3.2). The study periods of VH of Subregions II, III, and IV were set to DOY 121–193, 121–205, and 97–169, respectively (Fig. 8). Second, since the distances calculated from different bands (SWIR1 and VH) were related to their values. SWIR1 is the reflectance and has a value ranging from 0 to 1, while VH is the backscattering coefficient and has a value ranging from −50 dB to 1 dB. Therefore, the distances calculated from these two bands are not comparable. In order to combine the distances of the two bands, the distance was replaced by the ranking of the pixel by sorting the distance. Specifically, the distance calculated from the two bands were sorted separately, and the ranking of pixels ranged from 1 to the total number of cropland pixels. Noticed that the area of a 20-m resolution SWIR1 pixel is equivalent to four 10-m resolution VH pixels. That means the total number of SWIR1 pixels is one fourth of VH. Therefore, the ranking of SWIR1 needed to be multiplied by four on each pixel and resampled to 10-m resolution. By following this process, the rankings of two bands would be comparable and the pixel sizes would correspond. Third, the combined dissimilarity of each pixel was calculated by a weighted sum of the rankings of two bands. Since a weighted sum has been used, the sum of the two weights should be equal to 1. Therefore, only the weight of SWIR1 needs to be set here, and the weight of VH can be calculated accordingly. In this study, the weight of SWIR1 was determined based on the quality of the optical images. Specifically, the times of good observations of the optical images were used to calculate the weight of SWIR1. Since the TWDTW method with translation

stretching was used, the times of good observations referred to the times of good observations during the period corresponding to the minimum TWDTW distance of SWIR1 (section 2.3.3). Since the weight $w$ needs to be between 0 and 1, a function is required to map the number of good observations to a value between 0 and 1. The logistic function is commonly used to perform this type of mapping in various studies. This function was used to calculate the time weights mentioned previously, and its special form, the sigmoid function, has also been utilized as an activation function in some artificial neural networks (Maus et al., 2016; Han and Moraga, 1995). The formula of the logistic function is:

$$w = \frac{1}{1 + e^{-\alpha(x-\beta)}} \tag{3}$$

where $x$ is the times of good observations and $\alpha$ and $\beta$ are parameters. Through a small range of tests, $\alpha$ and $\beta$ were set to 2 and 2.5, respectively. By setting the parameters, $w$ was close to 1 when $x$ was greater than 3, and close to 0 when $x$ was less than 2 (Fig. 9). A higher weight would give to VH only in the case of very poor optical observations.

The combined dissimilarity $d$ was calculated as:

$$d = r_{SWIR1} \times w + r_{VH} \times (1 - w) \tag{4}$$

where $r_{SWIR1}$ and $r_{VH}$ are the ranking of SWIR1 and VH, respectively.

The distribution map was generated from the combined dissimilarity using the threshold mentioned in section 2.3.1.

### 2.3.5 Accuracy assessment

The study assessed the accuracy of the distribution map by using field data and county-level statistical areas. In this study, the confusion matrix was used to show the classification of the distribution map on the survey samples, and three accuracies were calculated, including the producer's accuracy (PA), user's accuracy, and overall accuracy (OA), calculated as:

$$PA = \frac{TP}{TP + FP} \times 100\ \% \tag{5}$$

$$UA = \frac{TP}{TP + FN} \times 100\ \% \tag{6}$$

$$OA = \frac{TP + TN}{TP + TN + FP + FN} \times 100\ \% \tag{7}$$

where $TP$ is the number of correctly classified single-season rice samples. $TN$ is the number of correctly classified non-single-season rice samples. $FP$ is the number of non-single-season rice samples classified as single-season rice. $FN$ is the number of single-season rice samples classified as non-single-season rice.

The county-level statistical planting areas from statistical year books were also used to verify the accuracy of the distribution map by comparing with the identified planting area at the county level. The relationships between the identified areas and the statistical areas were evaluated by linear regression. The coefficient of determination ($R^2$) and a relative error

(RMAE) are calculated. The calculation equation of RMAE is as follows:

$$\text{RMAE} = \frac{\sum_{i=1}^{n} |SA_i - IA_i|}{\sum_{i=1}^{n} SA_i} \tag{8}$$

where $SA_i$ and $IA_i$ are the statistical area and identified area of the $i$th county, and n indicates the number of counties.

## 3. Results

This study generated the distribution maps of single-season rice from 2017 to 2022 in 21 provincial administrative regions in China, which well reproduced the distribution of single-season rice in China (Fig. 10). Northeast China Plain, Yangtze Plain, and Sichuan Basin are three major single-season rice production areas in China, and single-season rice is planted most frequently in Northeast China Plain (Fig. 10). To show the ability of the distribution map of representing the details of rice fields, we chose three UAV sites and compared the distribution map with very-high-resolution UAV images (Fig. 11). Figures 11a and 11c were taken in July and show single-season rice fields in dark green (light green areas represent other planted vegetation). Figure 11b was taken in October, when single-season rice was about to be harvested, showing single-season rice fields in yellow-green. Despite some noise, the single-season rice fields were well classified in our distribution map (Fig. 11d–f).

The distribution map shows good performance in most of the provincial administrative regions. On average, the user's, producer's, and overall accuracy over all 21 provincial administrative regions were 73.08 %, 82.81 %, and 85.23 %, respectively (Table 2). The average overall accuracies in the four subregions were 95.69 %, 81.15 %, 86.75 %, and 80.18 %, respectively (Table 2). Subregion I (i.e., the northern provinces), had higher accuracy, while the southern provinces, especially the provinces in subregion IV (southwest) had poor accuracy. User's and producer's accuracies varied more between provinces than overall accuracy. The best user's and producer's accuracy all appeared in the northern provinces; the best user's accuracy was obtained for Inner Mongolia (97.67 %), and the best producer's accuracy for Liaoning (99.75 %) (Table 2). The lowest user's accuracy appeared in Guangxi (46.96 %), and the lowest producer's accuracy appeared in Jiangxi (49.22 %) (Table 2).

The county-level comparison with the statistical data showed a good performance. The identified area and statistical area had a very strong correlation, and the regression line was very close to the 1:1 line over all three years (Fig. 12). The slope ranged from 0.86 to 0.90, and $R^2$ ranged from 0.78 to 0.86.

Comparing on each province, the $R^2$ of the distribution map compared with the statistical data ranged from 0.15 to 0.94; the slope ranged from 0.24 to 1.44, and the relative error ranged from 0.24 to 0.56 (Fig. 13). Subregion I had the highest accuracy with an average $R^2$ of 0.92, followed by Subregion II with an average $R^2$ of 0.70, and subregions III and IV had poorer precision, both with an average $R^2$ of 0.55. Several provinces with more mountainous areas (Fujian, Guangxi, and Guizhou) had lower accuracies, while plain and main production provinces (Jilin, Liaoning, Jiangsu, Anhui, and Hubei) had higher

accuracies.

## 4. Discussion

Paddy rice is the second most widely planted crop in China. Its planting area has been relatively stable for many years (*China Statistical Yearbook*, 2020). From 1978 to 2005, the planting area of paddy rice decreased slowly, from 34.4 million hectares to 28.9 million hectares, and the planting area has been maintained at about 30 million hectares after 2005 (*China Statistical Yearbook*, 2020). The planting area of single-season rice accounted for two-thirds of all rice in China, and the production accounted for three-quarters of all rice (https://data.stats.gov.cn). Despite the importance of the single-season rice, rice mapping on a regional scale is still difficult.

### 4.1 Advantages of the TWDTW method

Many efforts have been made to map rice with a moderate or high spatial resolution at the provincial and regional scale, using machine learning methods and phenology-based methods (Pan et al., 2021; Xiao et al., 2006, 2005; You et al., 2021). However, these mapping methods have some limitations. Compared to machine learning methods, the TWDTW method has the advantage of requiring fewer training samples. In this study, the number of samples used in obtaining the standard time series was only 50. Many studies have reported that using machine learning methods to achieve high accuracy requires a large volume of training samples while obtaining samples is time-consuming and labor-intensive (Millard and Richardson, 2015; Valero et al., 2016). Therefore, the TWDTW method can be easily extended to regions or years with limited survey data compared to the machine learning methods. For example, You et al. (2021) mapped three crops, including rice in Northeast China from 2017 to 2020 by using a machine learning method (Random Forest), and achieved producer's and user's accuracy of rice greater than 90 %, except for the user's accuracy in 2017 (87 %). However, their study used more than 8000 training samples per year that needed to be updated every year. In contrast, this study achieved a similar accuracy with only 50 sample points per province in Northeast China.

Compared with the flood-detection method developed by Xiao et al. (2005), the TWDTW method uses signals in a certain period before and after rice flooding, including more phenological information. Flood-detection methods are deeply affected by clouds and rain. The accuracy of a moderate-resolution rice map based on the MODIS data can be relatively high due to the high temporal resolution and less cloudy pixels of the MODIS data (Xiao et al., 2006, 2005). However, when based on Landsat data, the accuracy of such high-resolution product was not satisfactory due to the influence of cloudy images (Dong et al., 2016). Furthermore, good observations in the southern areas of China are extremely scarce, especially in Subregion IV (southwestern), where the six-year average of the frequency of good observations is only between 25 % and 40 % during the

transplanting period, making it impossible to map rice in these provinces using only optical images (Fig. 2). SAR images were introduced in this study in cloudy areas, making it possible to map rice in these areas.

## 4.2  Uncertainty analysis

The introduction of SAR has made rice identification possible in these areas. However, the quality of SAR images is somewhat worse than that of optical images, which makes the accuracy of the distribution map in these areas still lower than that in less cloudy areas. The optical data for 2017 have the poorest observation quality, with the number of observations corresponding to the minimum distance only 1.16 (Fig. 14). This number of observations determines the high weight of the distances calculated from the SAR images, which explains the lowest $R^2$ in 2017 (Fig. 12). Comparing the $R^2$ of the county-level comparison with the statistical data with the frequencies of optical observations during the study period, shows a positive correlation with $R^2$ range from 0.35 to 0.57 (Fig. 15). That is, in areas where the optical observations are heavily affected by clouds, the accuracy remains low to some extent, even if SAR images are used as auxiliaries.

Another important factor that affects identification accuracy is the fragmentation of planted areas. In mountainous provinces such as Guizhou, Chongqing, and Yunnan, there are many terraced rice fields, which are very narrow and fragmented (Cao et al., 2021; Yan et al., 2016). In these mountainous areas, rice fields may be less than 10 m wide, resulting in mixed pixels at 10-m resolution. In addition, as a side-looking radar system, SAR has a terrain effect, which produces more errors in mountainous areas (Beaudoin et al., 1995). To quantify the fragmentation of the distribution map, we regarded adjacent single-season rice pixels as a patch, and counted the size of each patch. The fragmentation of the distribution maps in the same province varied little from year to year (Fig. 16). Guangxi, Guizhou, Shaanxi, Yunnan, and Fujian were the most fragmented provinces, with more than half of the pixels belonging to patches smaller than 100 pixels (about 1 hectare). The most fragmented province was Guangxi, where, each year, an average of 85.45 % of the pixels belonged to patches smaller than 100 pixels (Fig. 16). Although there are plains in Guangxi, the plains are mostly planted with double-season rice, while single-season rice is mostly planted in mountainous areas, resulting in extremely fragmented single-season rice cultivation. Using the percentage of pixels belonging to patches smaller than 100 pixels as an indicator of fragmentation, and comparing it with the identification accuracy, a significant negative correlation can be found (Fig. 17). The $R^2$ of the fragmentation and identification accuracy ranged from 0.51 to 0.72, confirming that the fragmentation of single-season rice fields has a strong negative effect on the identification accuracy and is an important source of identification error (Fig. 17).

In recent years, due to the shortage of rural labor, direct-seeded rice (DSR) has been increasingly used in China (Chakraborty et al., 2017). Unlike transplanting, DSR does not require seedling raising and transplanting. Instead, the seeds are sown directly in the main field. Depending on the field conditions, there are three types of DSR: wet direct seeding, water

direct seeding and dry direct seeding (Farooq et al., 2011). The wet direct seeding sowed the seed in puddled soil surface, and the water direct seeding sowed the seed in flooded fields. Most of the DSR belongs to these two types. In contrast, the dry direct seeding sowed the seed in a dry field. Therefore, our method can be used to identify rice fields of wet or water direct seeding by capturing the moisture or flood signal, while rice fields using dry direct seeding cannot be identified using our method. Some studies have also pointed out that certain types of DSR may have a weak flooding signal compared to

transplanting, making it difficult to distinguish them from other crops using traditional classification methods (Guo et al., 2019). At present, the proportion of dry direct seeding in China is small, and it has a limited impact on the accuracy of the distribution map. However, as dry direct seeding continues to spread, its impact on rice mapping will become difficult to ignore. New methods for rice mapping must be developed in the future.

### 4.3    Future development

Rice mapping strongly depends on the distinctive spectral characteristics of the flooding period. Its spectral characteristics in the growing and harvesting periods are very similar to those of other summer crops. Therefore, previous studies all chose to capture the characteristics of the flooding period. However, optical and SAR images have their own limits during this short period. In this study, we combined two sources of satellite images together to overcome the limitations of each source of satellite data. However, this combination was still relatively simple. Some recent data fusion studies use machine learning

methods to reconstruct high-quality optical data with both high spatial and temporal resolutions. We hope that these kinds of reconstructed datasets will help solve the limitations of optical images and help to produce more accurate single-season rice maps.

### 5.    Data availability

The distribution map of single-season rice of 21 provincial administrative regions in China from 2017 to 2022 is available

at https://doi.org/10.57760/sciencedb.06963 (Shen et al., 2023). The file format of the product is GeoTIFF with the spatial reference of WGS84 (EPSG:4326). The distribution map of single-season rice will be updated annually at the end of each year.

### 6.    Conclusions

This study proposed a new rice mapping method based on the time-weighted dynamic time warping (TWDTW) method. The TWDTW distances of the shortwave infrared1 (SWIR1) band from optical images and of the VH band from synthetic

aperture radar (SAR) images were combined according to a weight, and the number of good optical observations was used to

determine the weight. By using this method, this study produced distribution maps of single-season rice in China from 2017 to 2022 at 10-m or 20-m resolution. The overall accuracy over 21 provincial administrative regions averaged 85.23 % based on 108195 samples; the average $R^2$ was 0.83 over three years compared with county-level statistical planting areas. However, the method did not fully resolve the limitations of optical and SAR images as clouds and the fragmentation of the rice fields

still affected the accuracy of the distribution map. In general, this study produced high-resolution single-season rice maps of China, and the method can be easily applied to other regions and the maps can be updated annually.

**Author contributions**

WY and RS designed the research. BP, RS, QP, JD, XC, XZ, TY, and JH performed the investigation. RS, BP, JD, and WY developed the method. RS implemented the computer code, performed the formal analysis, visualized the results, and

wrote the manuscript. WY and QP edited and revised the manuscript.

**Competing interests**

The authors declare that they have no conflict of interest.

**Acknowledgements**

We would like to thank the editors and reviewers for their constructive comments.

**Financial support**

This study was supported by the Open Research Program of the International Research Center of Big Data for Sustainable Development Goals (Grant No. CBAS2023ORP02).

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

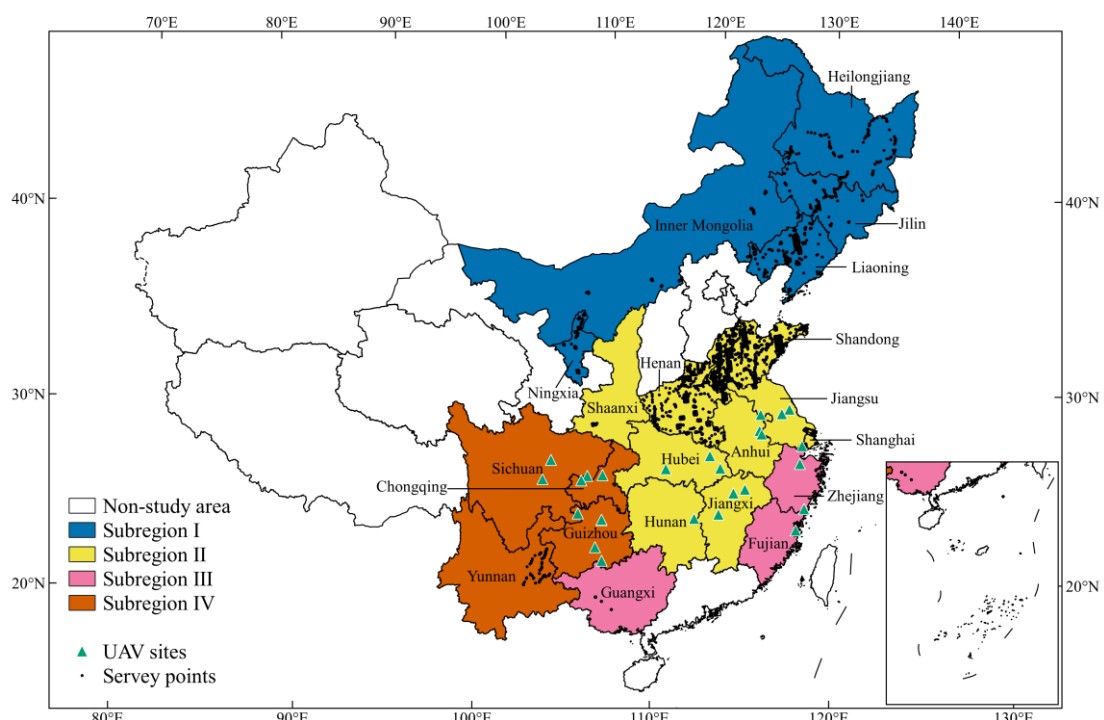

**Figure 1:** The study area includes 21 provincial administrative regions in China and is divided into four subregions (colored areas). The black dots indicate the samples obtained from the survey, and the green triangles indicate the unmanned aerial 490 vehicle (UAV) survey sites.

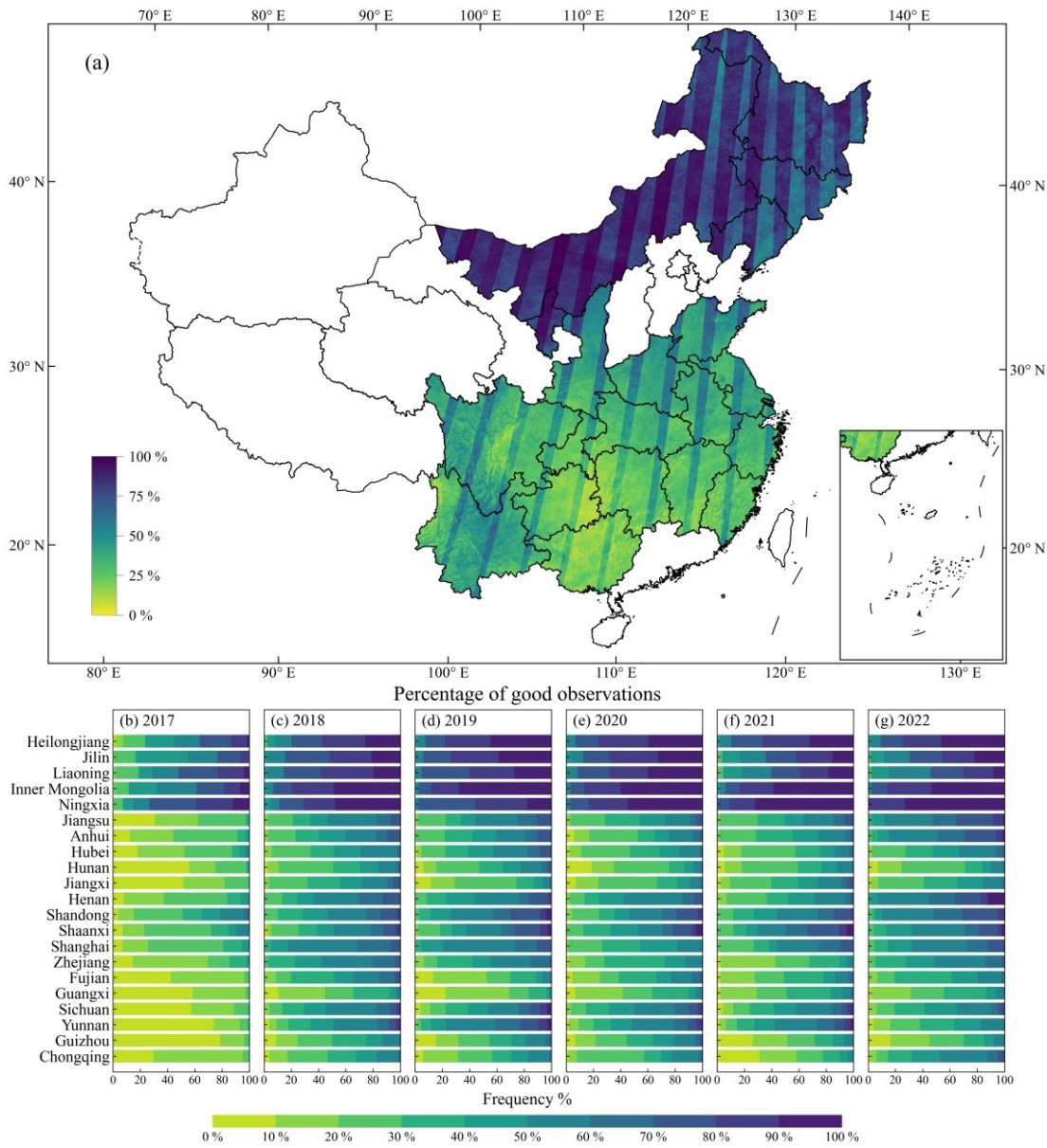

**Figure 2:** Percentage of good observations of Sentinel-2 during the study period of 2017–2022 (a). The bottom row (b–g) shows the frequencies of percentages of good observations in each province during the study period of each year.


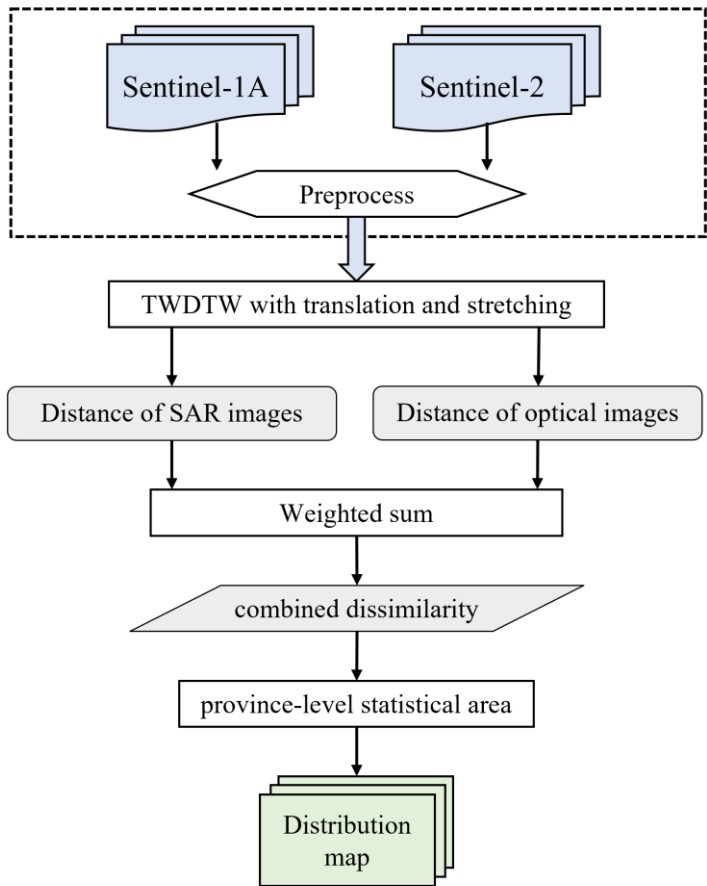

**Figure 3:** The conceptual flow chart of the method.

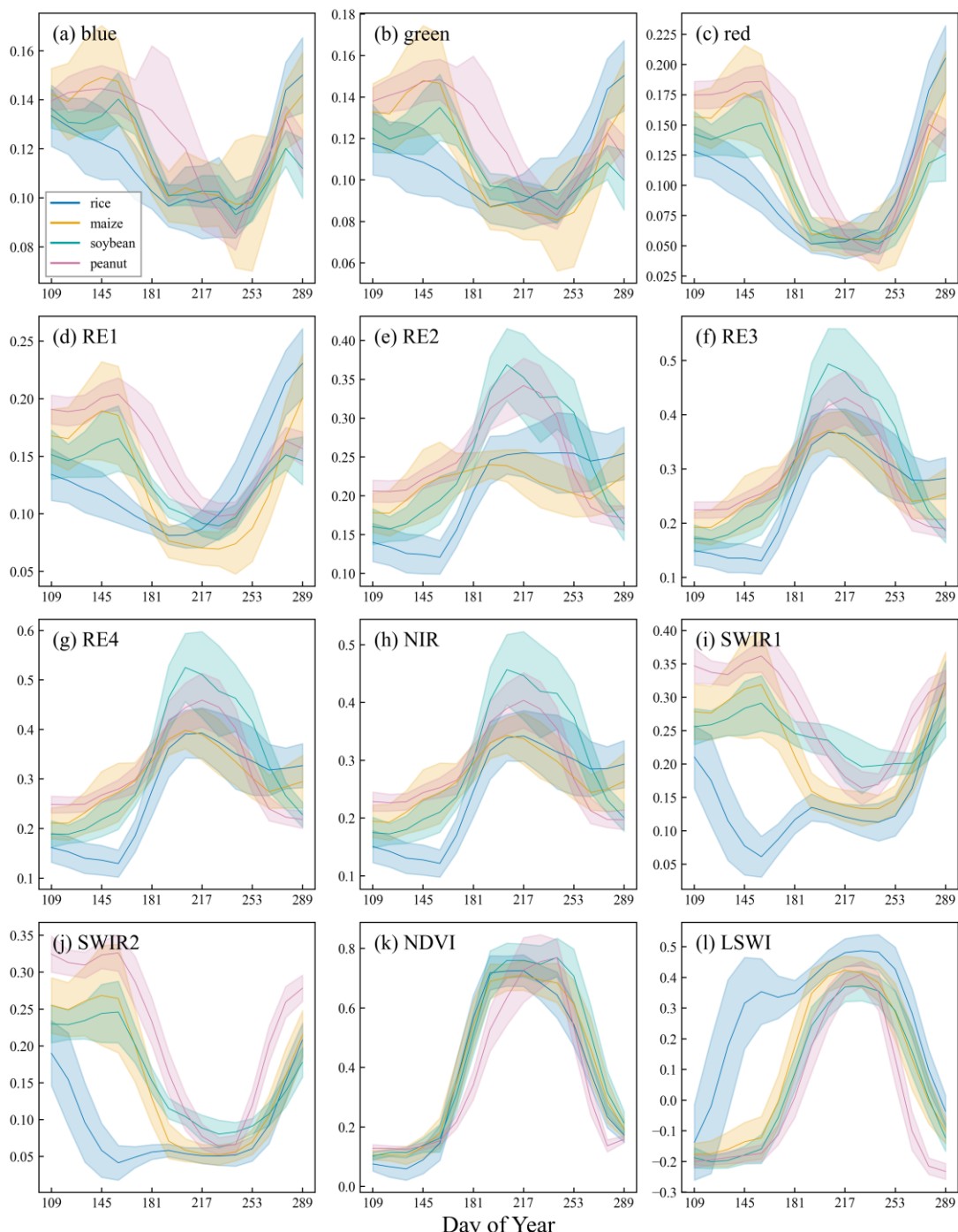

**Figure 4:** Time series of 10 optical bands or indexes over four main crop types in Jilin province in 2019. Solid lines indicate

the average time series, and the shaded error bands represent the standard deviations.

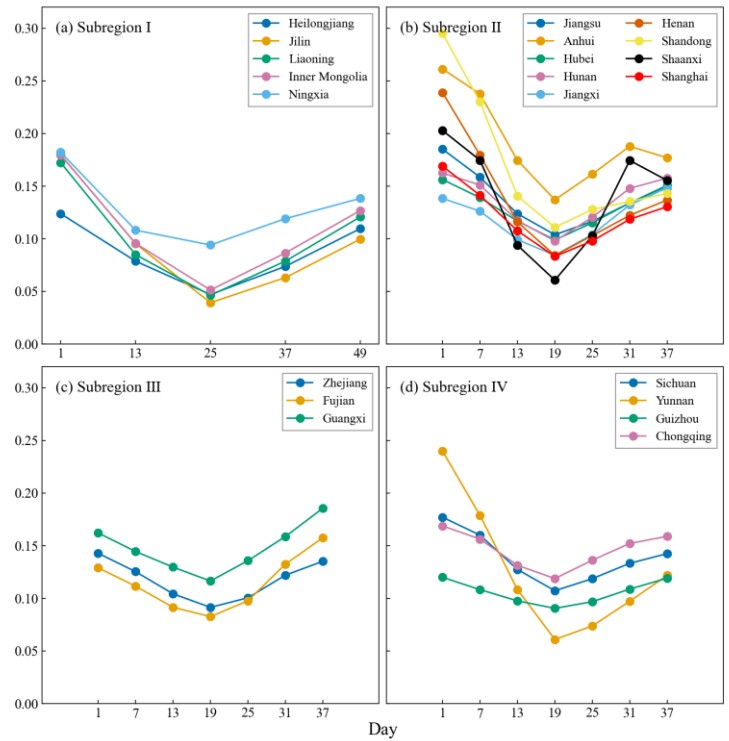

**Figure 5:** Standard SWIR1 time series of single-season rice in 21 provincial administrative regions in four subregions.

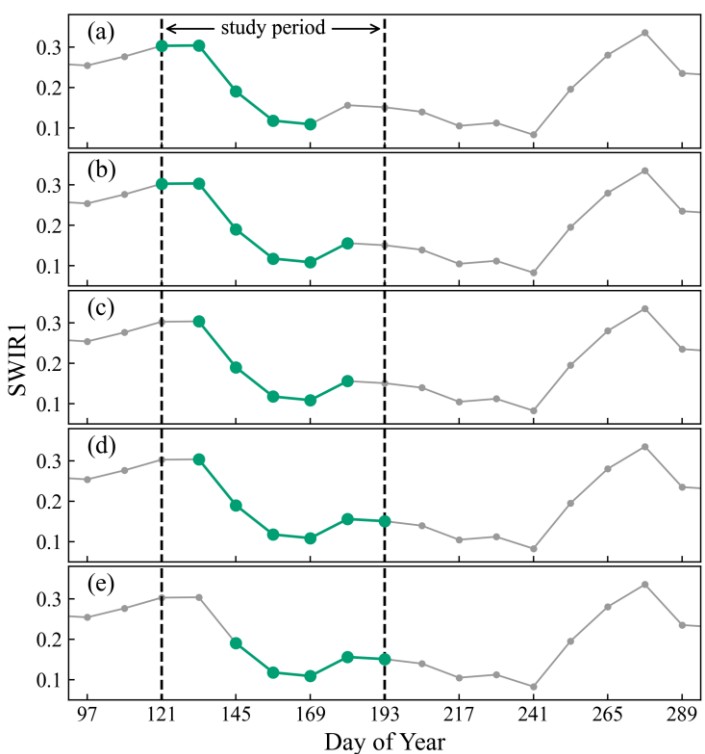


**Figure 6:** A time series of SWIR1 in Jilin Province. The period between dashed lines (DOY 121–193) is the study period. Five green curves are the time series selected to calculate the distance with the standard time series using the TWDTW method.

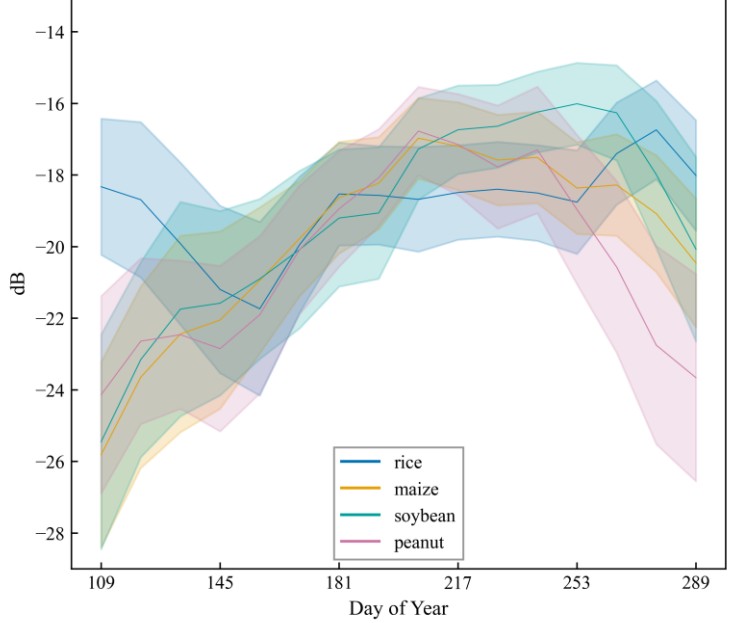


**Figure 7:** Time series of the VH band over four main crop types in Henan Province in 2019. Solid lines indicate the average

time series; the shaded error bands represent the standard deviations.

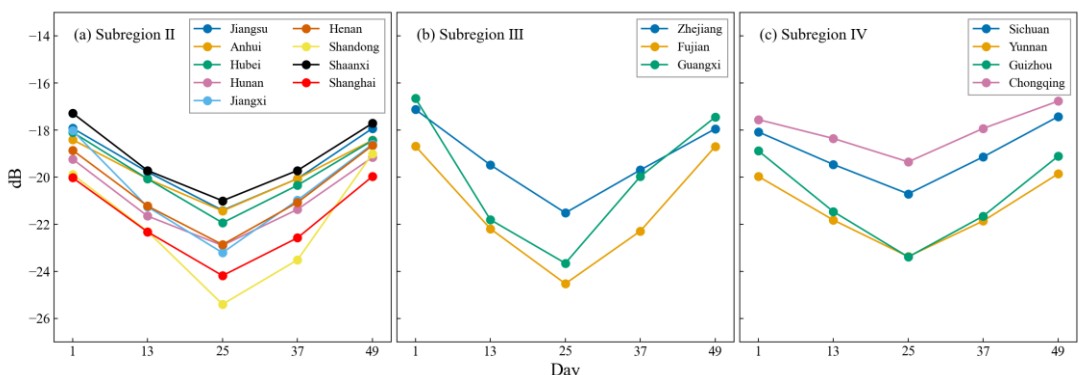

**Figure 8:** Standard VH time series of single-season rice in 16 provincial administrative regions in Subregions II, III, and IV.

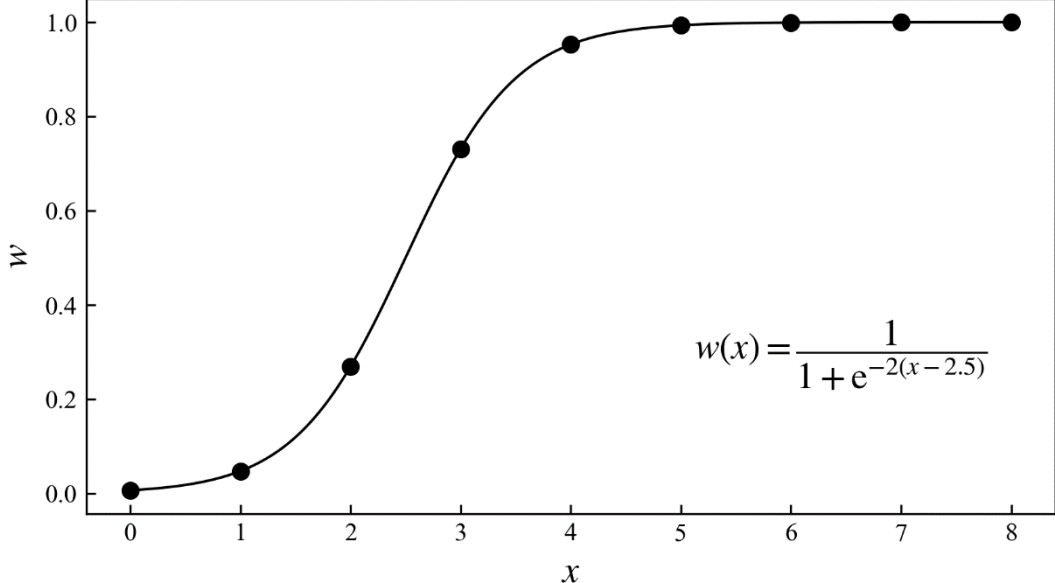

**Figure 9:** Times of good observation *x* and the corresponding weights *w* of SWIR1.

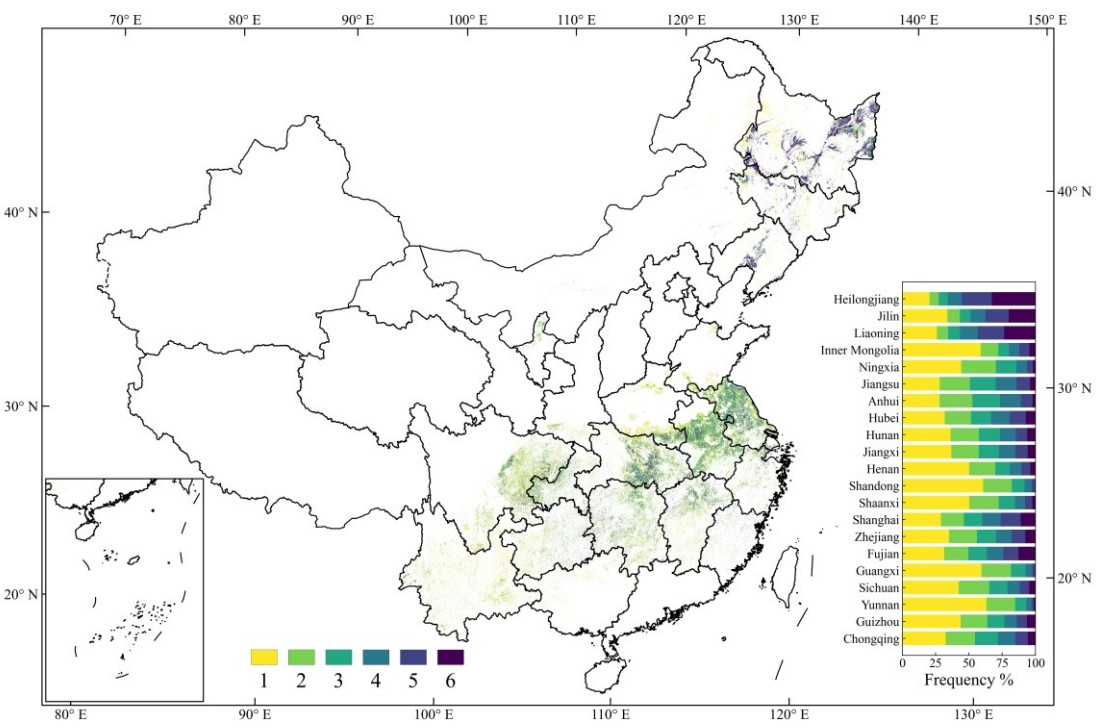


**Figure 10:** Planting frequency of single-season rice in China from 2017 to 2022.

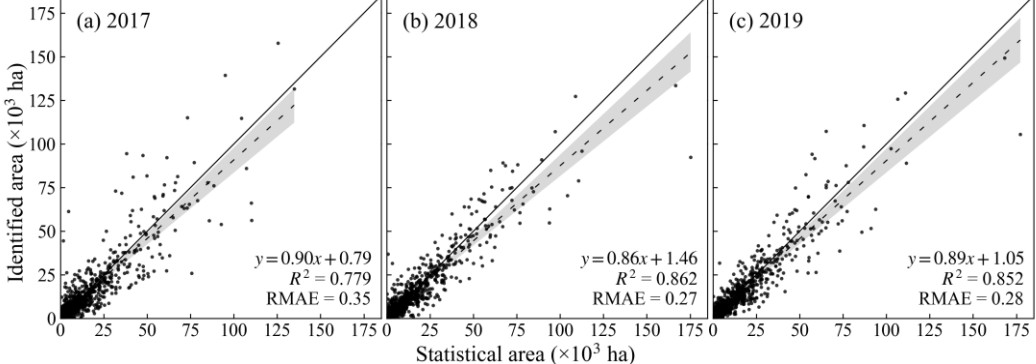

**Figure 11:** Distribution map in three UAV sites of Hubei (114°47′49″ E, 31°1′11″ N), Zhejiang (120°32′33″ E, 29°57′14″ N),

and Sichuan (106°44′15″ E, 30°19′5″ N). (a)–(c) are very-high-resolution UAV images taken at three sites on July 8, 2018,

October 12, 2018, and July 13, 2018, respectively. (d)–(f) overlaid distribution maps with identified single-season rice pixels

indicated in red.

**Figure 12:** County-level comparison of identified and statistical planting areas of 2017–2019. Solid lines are 1:1 lines; dashed

lines are regression lines. The confidence intervals are shaded in gray.

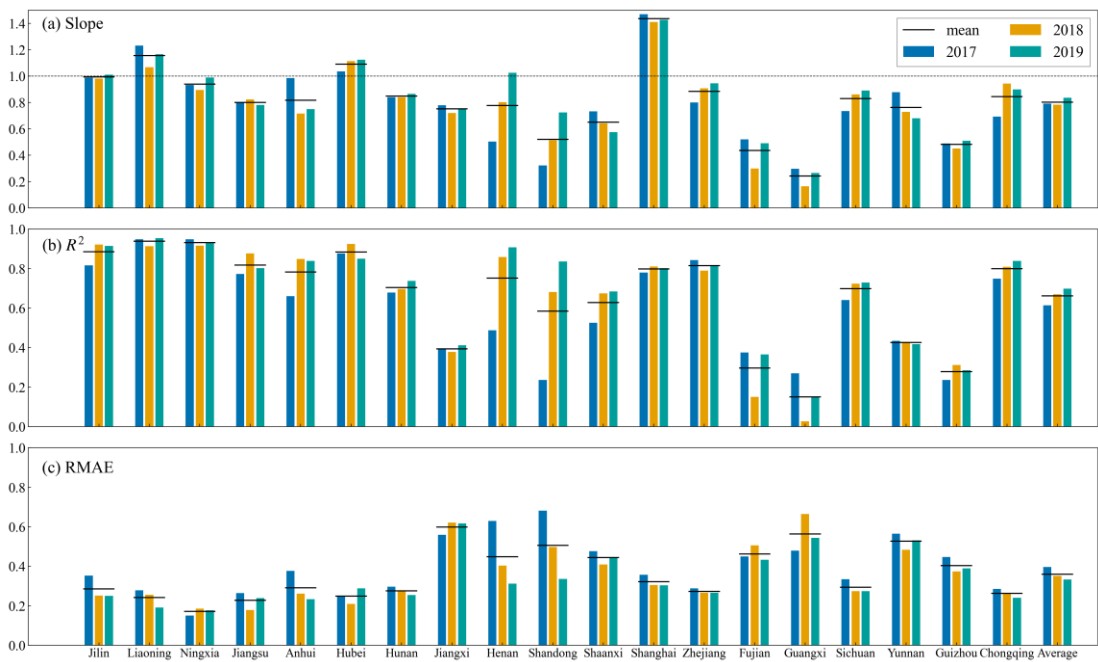

**Figure 13:** Comparison between identified and statistical planting areas at the county-level of 2017–2020 in 19 provincial

administrative regions.

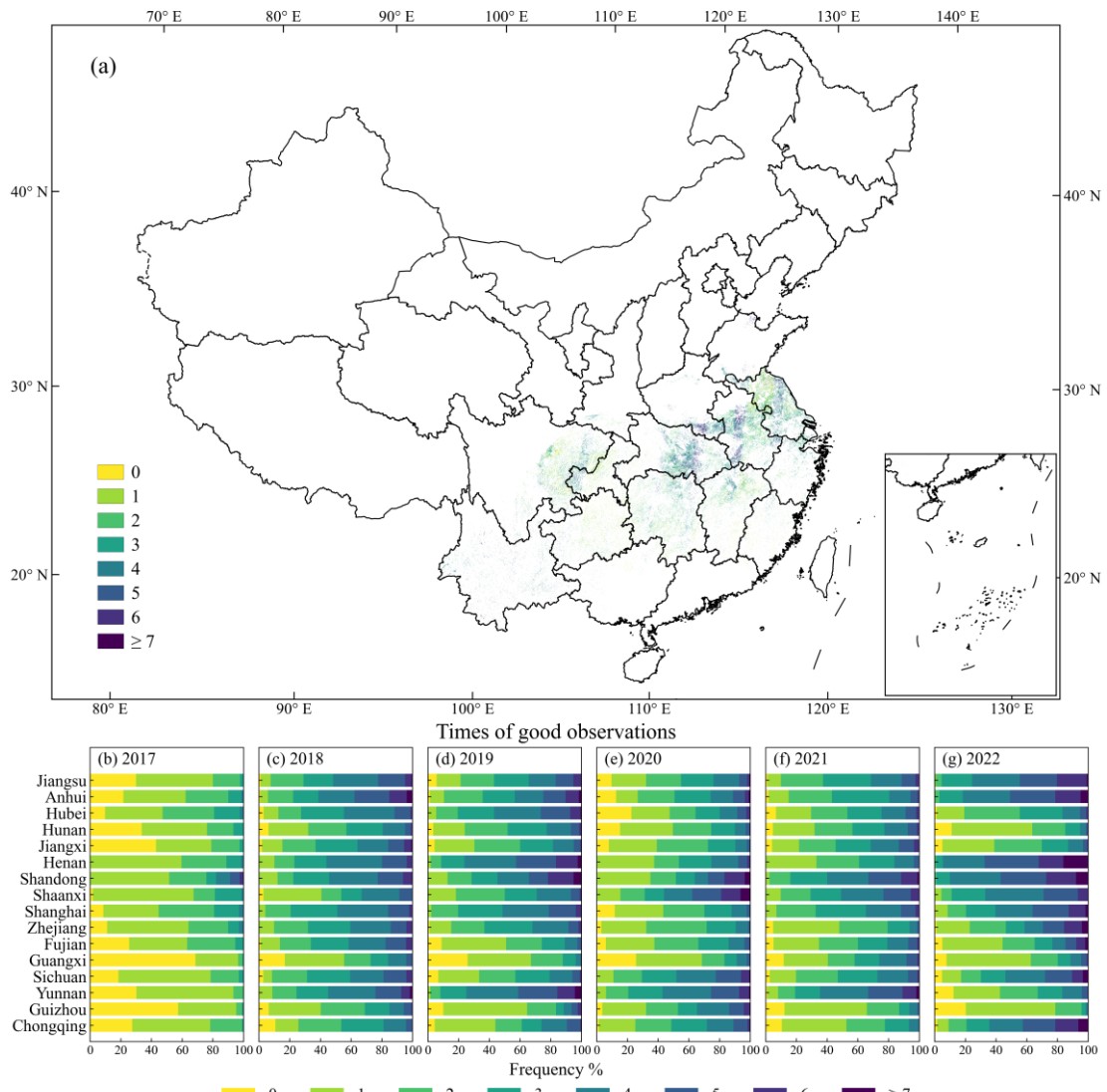

**Figure 14:** Times of good observations of Sentinel-2 during the time period corresponding to the minimum TWDTW distance in identified single-season rice pixels in 19 provincial administrative regions in Subregion II, III, and IV in 2019. The bottom row (b–g) shows the times of good observations in each province during the time period corresponding to the minimum TWDTW distance of each year.

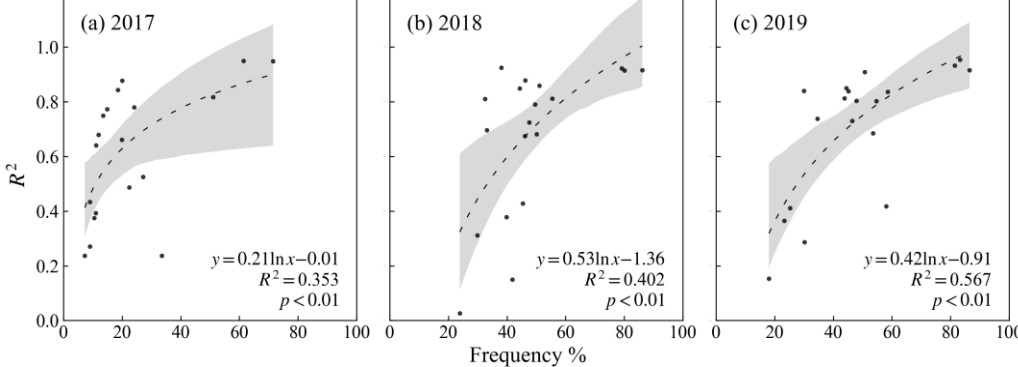

**Figure 15:** Relationship between identification accuracies ($R^2$ of county-level comparison with the statistical planting area) and provincial mean of good observation frequencies in 19 provincial administrative regions from 2017 to 2020. Dashed lines

are regression lines; the confidence intervals are shaded in gray.

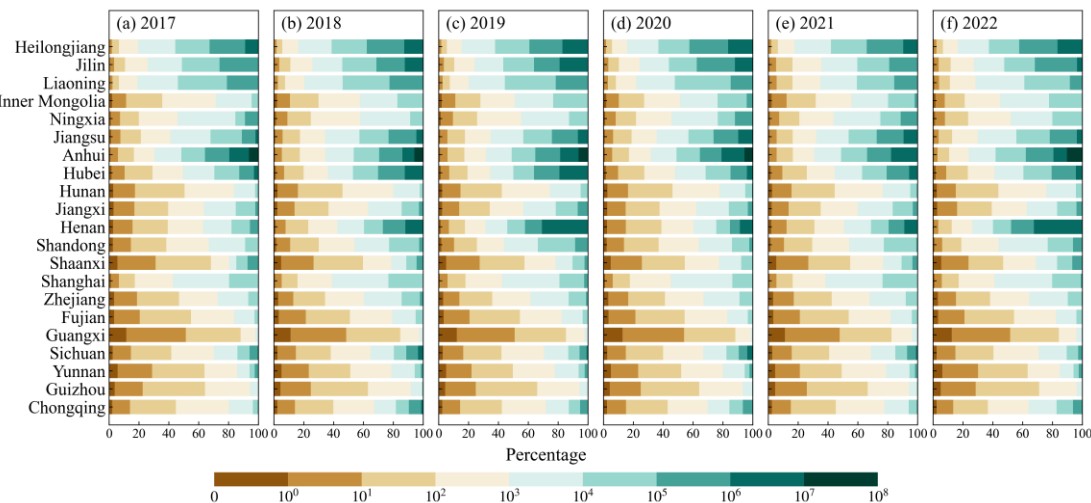

**Figure 16:** Distribution of the number of single patches.

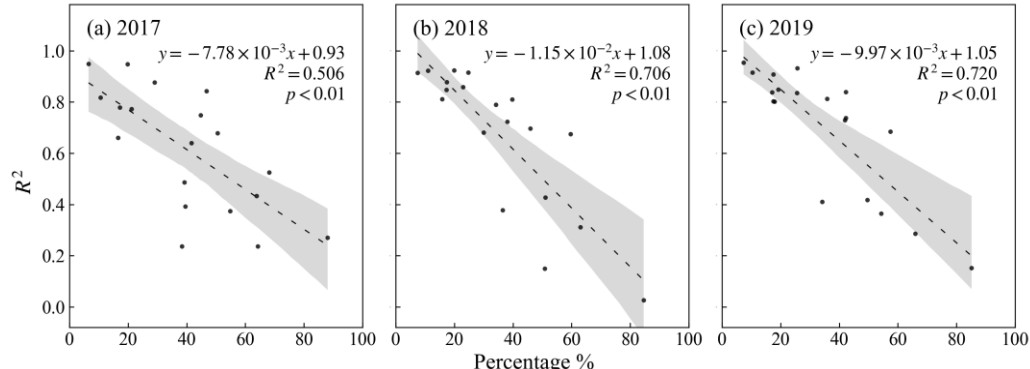

**Figure 17:** Relationship between identification accuracies ($R^2$ of county-level comparison with the statistical planting area) and percentage of patches with a size less than or equal to 100 in 19 provincial administrative regions from 2017 to 2020. Dashed lines are regression lines; the confidence intervals are shaded in gray.

**Table 1:** Number of available county-level statistical data

| Province | Total number of counties | 2017 | 2018 | 2019 |
|---|---|---|---|---|
| Heilongjiang | 128 | 0 | 0 | 0 |
| Jilin | 60 | 49 | 50 | 50 |
| Liaoning | 100 | 20 | 21 | 21 |
| Inner Mongolia | 102 | 0 | 0 | 0 |
| Ningxia | 22 | 10 | 10 | 10 |
| Jiangsu | 99 | 66 | 71 | 60 |
| Anhui | 104 | 57 | 90 | 68 |
| Hubei | 103 | 78 | 78 | 77 |
| Hunan | 122 | 111 | 119 | 117 |

| Jiangxi | 100 | 23 | 22 | 22 |
|---|---|---|---|---|
| Henan | 158 | 39 | 50 | 38 |
| Shandong | 137 | 18 | 18 | 18 |
| Shaanxi | 107 | 21 | 21 | 21 |
| Shanghai | 17 | 9 | 9 | 9 |
| Zhejiang | 90 | 22 | 17 | 22 |
| Fujian | 85 | 24 | 19 | 20 |
| Guangxi | 110 | 44 | 29 | 27 |
| Sichuan | 183 | 95 | 102 | 102 |
| Yunnan | 129 | 125 | 125 | 125 |
| Guizhou | 88 | 56 | 56 | 56 |
| Chongqing | 38 | 37 | 37 | 37 |

**Table 2:** Confusion matrices of the distribution map in 21 provincial administrative regions.

| Province | Class | SR[1] | Other | UA (%) | PA (%) | OA (%) |
|---|---|---|---|---|---|---|
| Heilongjiang | SR[2] | 164 | 5 | 89.13 | 97.04 | 95.70 |
| | Other | 20 | 393 | 98.74 | 95.16 | |
| Jilin | SR | 5598 | 16 | 90.32 | 99.71 | 96.77 |
| | Other | 600 | 12840 | 99.88 | 95.54 | |
| Liaoning | SR | 5890 | 15 | 92.41 | 99.75 | 96.87 |
| | Other | 484 | 9541 | 99.84 | 95.17 | |
| Inner Mongolia | SR | 84 | 10 | 97.67 | 89.36 | 98.09 |
| | Other | 2 | 531 | 98.15 | 99.62 | |
| Ningxia | SR | 47 | 5 | 69.12 | 90.38 | 91.03 |
| | Other | 21 | 217 | 97.75 | 91.18 | |
| Jiangsu | SR | 2249 | 58 | 62.42 | 97.49 | 67.14 |
| | Other | 1354 | 636 | 91.64 | 31.96 | |
| Anhui | SR | 1133 | 168 | 54.55 | 87.09 | 63.88 |
| | Other | 944 | 834 | 83.23 | 46.91 | |
| Hubei | SR | 2034 | 206 | 87.18 | 90.8 | 87.31 |
| | Other | 299 | 1441 | 87.49 | 82.82 | |
| Hunan | SR | 397 | 62 | 62.92 | 86.49 | 83.77 |
| | Other | 234 | 1131 | 94.80 | 82.86 | |
| Jiangxi | SR | 603 | 622 | 70.2 | 49.22 | 67.18 |
| | Other | 256 | 1194 | 65.75 | 82.34 | |
| Henan | SR | 2694 | 57 | 95.80 | 97.93 | 99.13 |
| | Other | 118 | 17315 | 99.67 | 99.32 | |
| Shandong | SR | 1977 | 241 | 72.47 | 89.13 | 89.60 |
| | Other | 751 | 6566 | 96.46 | 89.74 | |
| Shaanxi | SR | 454 | 43 | 71.50 | 91.35 | 84.18 |
| | Other | 181 | 738 | 94.49 | 80.30 | |
| Shanghai | SR | 128 | 7 | 83.12 | 94.81 | 88.13 |

| | | | | | | |
|---|---|---|---|---|---|---|
| | Other | 26 | 117 | 94.35 | 81.82 | |
| Zhejiang | SR | 900 | 200 | 85.88 | 81.82 | 90.67 |
| | Other | 148 | 2480 | 92.54 | 94.37 | |
| Fujian | SR | 530 | 108 | 94.81 | 83.07 | 90.36 |
| | Other | 29 | 754 | 87.47 | 96.30 | |
| Guangxi | SR | 108 | 25 | 46.96 | 81.2 | 79.21 |
| | Other | 122 | 452 | 94.76 | 78.75 | |
| Sichuan | SR | 2031 | 353 | 62.51 | 85.19 | 77.16 |
| | Other | 1218 | 3275 | 90.27 | 72.89 | |
| Yunnan | SR | 78 | 72 | 69.64 | 52.00 | 88.87 |
| | Other | 34 | 768 | 91.43 | 95.76 | |
| Guizhou | SR | 1836 | 477 | 83.42 | 79.38 | 89.39 |
| | Other | 365 | 5257 | 91.68 | 93.51 | |
| Chongqing | SR | 486 | 408 | 54.30 | 54.36 | 71.07 |
| | Other | 409 | 1521 | 78.85 | 78.81 | |

[1]number of field surveyed samples. [2]number of identified samples. SR is single-season rice.

560