# Peer review of "High-resolution distribution maps of single-season rice in China from 2017 to 2022"

_Earth System Science Data, 2023_

## Author Comment (AC1)

**Reply to Referee #1**

**Food security is crucial to human survival, and this article's proposed large-scale fine-resolution single-season rice mapping method is meaningful. However, there are some questions or problems:**

Thanks for your positive comments. We revised the manuscript based on the comments.

**1. This article uses SAR data in multi-cloud pixels, which is indeed not affected by clouds and mist. However, SAR images are affected by salt-and-pepper noise, can filtering or homogeneous sample point selection method be considered for denoising?**

Sorry for the confusion. The SG filter method mentioned in section 2.2.1 was applied to both Sentinel-2 and Sentinel-1 time series. We have revised the sentence to clarify it.

"To further eliminate the noise in the time series of Sentinel-1 and Sentinel-2 images, a Savitzky-Golay (SG) filter with the order set to two and the window size set to five was applied to smooth the time series." (Lines 108–110)

**2. The compatibility issue between the SAR VH band and the optical image's SWIR1 band needs to be solved, and a clearer explanation is needed. How do you prove that your processing method regarding this is feasible?**

We have revised the section and added several sentences to explain it more clearly.

"Second, since the distances calculated from different bands (SWIR1 and VH) were related to their values. SWIR1 is the reflectance and has a value ranging from 0 to 1, while VH is the backscattering coefficient and has a value ranging from −50 dB to 1 dB. Therefore, the distances calculated from these two bands are not comparable. In order to combine the distances of the two bands, the distance was replaced by the ranking of the pixel by sorting the distance. Specifically, the distance calculated from the two bands were sorted separately, and the ranking of pixels ranged from 1 to the total number of cropland pixels. Noticed that the area of a 20-m resolution SWIR1

pixel is equivalent to four 10-m resolution VH pixels. That means the total number of SWIR1 pixels is one fourth of VH. Therefore, the ranking of SWIR1 needed to be multiplied by four on each pixel and resampled to 10-m resolution. By following this process, the rankings of two bands would be comparable and the pixel sizes would correspond." (Lines 182–190)

**3. How does this article handle data of different resolutions (such as 10m and 20m)?**

Please refer to the response #2. The ranking of SWIR1 would be multiplied by four on each pixel and resampled to 10-m resolution to ensure the size of pixels correspond.

**4. Are formulas 3 and 4 referenced? The explanation of formula 3 and its parameters should be more specific.**

$$w = \frac{1}{1 + e^{-\alpha(x-\beta)}} \tag{3}$$

$$d = r_{SWIR1} \times w + r_{VH} \times (1 - w) \tag{4}$$

There are no references to Formulas 3 and 4. It appears that no previous study has made a similar attempt in the field of crop mapping. We have added a few sentences and a new figure to clarify the design concept behind the formulas.

"Since a weighted sum has been used, the sum of the two weights should be equal to 1. Therefore, only the weight of SWIR1 needs to be set here, and the weight of VH can be calculated accordingly. In this study, the weight of SWIR1 was determined based on the quality of the optical images. Specifically, the times of good observations of the optical images were used to calculate the weight of SWIR1. Since the TWDTW method with translation stretching was used, the times of good observations referred to the times of good observations during the period corresponding to the minimum TWDTW distance of SWIR1 (section 2.3.3). Since the weight $w$ needs to be between 0 and 1, a function is required to map the number of good observations to a value between 0 and 1. The logistic function is commonly used to perform this type of mapping in various studies. This function was used to calculate the time weights mentioned previously, and its special form, the sigmoid function, has

also been utilized as an activation function in some artificial neural networks (Maus et al., 2016; Han and Moraga, 1995). The formula of the logistic function is:

$$w = \frac{1}{1 + e^{-\alpha(x-\beta)}} \qquad (3)$$

where $x$ is the times of good observations and $\alpha$ and $\beta$ are parameters. Through a small range of tests, $\alpha$ and $\beta$ were set to 2 and 2.5, respectively. The length of the standard time series in subregion II, III and IV was 7, so the value of the times of good observations $x$ ranged from 0 to 8. By setting the parameters, $w$ was close to 1 when $x$ was greater than 3, and close to 0 when $x$ was less than 2 (Fig. 8). A higher weight would give to VH only in the case of very poor optical observations." (Lines 191–204)

[Figure]

$$w(x) = \frac{1}{1 + e^{-2(x-2.5)}}$$

**Figure 8: Times of good observation $x$ and the corresponding weights $w$ of SWIR1.**

**5. In line 217, figure 8a and 8c is not exist, this issue needs to be thoroughly checked.**

Sorry for not checking the image numbers carefully, these should be Figures 9a and 9c. We have rechecked all image numbers.

**6. The distribution of statistical data in 2017 is different from that in other years. Is it a problem of data processing?**

[Figure]

**Figure 11: County-level comparison of identified and statistical planting areas of 2017–2019. Solid lines are 1:1 lines; dashed lines are regression lines. The confidence intervals are shaded in gray.**

In this figure, the county with the largest statistical planting area was around 175 kha in 2018 and 2019, and less than 150 kha in 2017. We checked the original statistical yearbooks, and the county with the largest planting area of single-season rice in all three years was Huoqiu County, Anhui Province. The statistical planting areas of Huoqiu County in the three years were 134.81, 175.26, and 177.24 kha, respectively. Additionally, there was also a significant increase in the statistical planting area in Shouxian County, Anhui Province from 2017 to 2018. The statistical planting areas of Shouxian County in the three years were 125.5, 166.53, and 168.13 kha, respectively. Therefore, the original data itself showed such discrepancies, and were not caused by data processing. On the other hand, the number of counties included in the statistical data collection was different for different years, which also contributed to the apparent difference in the distribution of the data. Moreover, the accuracy of the planting map in 2017 was lower than that in the other two years, which could make the distribution look more scattered. We are aware of the uncertainty in statistical data and only use it for comparison purposes here, as there are no other options.

**Reply to Referee #2**

**This manuscript introduces a set of single-season rice map data in China, which has potential implications for food security and agricultural planning. The method and results are well presented. The data described in the paper is accessible and reusable. However, there are several points that require further clarification:**

Thanks for your positive comments. We revised the manuscript based on the comments.

**1. The study area encompasses 21 provinces in China. Could the authors elaborate on why other regions, such as the Northwestern region, were not included in the study?**

**"This study was conducted in 21 provincial administrative regions in mainland China, where the total planting area of single-season rice was 19.92 million hectares, accounting for approximately 99.01 % of the total planting area of single-season rice in mainland China according to the statistical data in 2018 (https://data.stats.gov.cn). The total production of the single-season rice in the study area was 150.46 million tons, accounting for approximately 98.91 % of the total production in mainland China in 2018." (Lines 72–76)**

The study area was selected based on statistical data of each provincial administrative region. The statistical planting area and production of single-season rice in the study area accounted for about 99 % of the total planting area and production of China. Almost no single-season rice is cultivated outside of the study area. Some Northwestern provinces, such as Qinghai, Gansu, and Xinjiang, have vast areas but very little rice cultivation. Including these provinces would significantly increase the calculation time. Therefore, only these 21 provincial administrative regions were selected as the study area in this study.

**2. The manuscript mentions a planting frequency map. Is this map available for download? I was unable to locate it in the provided data repository.**

The dataset does not include a planting frequency map, as this map is simply an overlay of distribution maps over the 6 years to provide an overview. We consider this to be an analysis and not a part of the data product. Therefore, it is not included in the dataset, however this map can be easily produced by using annual distribution datasets over the 6 years which have been provided by this manuscript.

**3. Figure 9 presents a comparison between the UAV image and the rice mapping result, but the comparison is hard to interpret. I recommend using the UAV image as the base map in subfigures d-f to facilitate clearer comparisons.**

We have revised the figure using UAV image as a base map.

[Figure]

**Figure 10:** Distribution map in three UAV sites of Hubei (114°47′49″ E, 31°1′11″ N), Zhejiang

(120°32′33″ E, 29°57′14″ N), and Sichuan (106°44′15″ E, 30°19′5″ N). (a)–(c) are very-high-resolution UAV images taken at three sites on July 8, 2018, October 12, 2018, and July 13, 2018, respectively. (d)–(f) overlaid distribution maps with identified single-season rice pixels indicated in red.

**4. The discussion section appears lengthy and unstructured. I suggest reorganizing this section and categorizing the discussion into subsections according to the topic.**

Thanks, we have divided the discussion section into three subsections, namely "Advantages of the TWDTW method", "Uncertainty analysis", and "Future development".

**5. Most importantly, the paper lacks a description of data sustainability, which is crucial for readers intending to reuse the data. The current data set includes the mapping result for 2017-2022. Are there plans to continue the project in subsequent years? It is unfortunate that many projects and data were discontinued after the paper was published. So I highly recommend including a data management plan in the manuscript, particularly given its submission to a scientific data journal. For instance, if the project is to continue, what is the operational plan? If not, how can users reproduce the data independently?**

Thanks for your constructive comment. Yes, it is very important to update the annual distribution, and we are willing to do so. We have added a sentence to the Data availability section to describe the data update plan.

"The distribution map of single-season rice will be updated annually at the end of each year." (Line 318)

**References**

Han, J. and Moraga, C.: The influence of the sigmoid function parameters on the speed of backpropagation learning, in: From Natural to Artificial Neural Computation, Berlin, Heidelberg, 195–201, https://doi.org/10.1007/3-540-59497-3_175, 1995.

Maus, V., Camara, G., Cartaxo, R., Sanchez, A., Ramos, F. M., and de Queiroz, G. R.: A Time-Weighted Dynamic Time Warping Method for Land-Use and Land-Cover

Mapping, IEEE Journal of Selected Topics in Applied Earth Observations and Remote Sensing, 9, 3729–3739, https://doi.org/10.1109/JSTARS.2016.2517118, 2016.

---

## Author Comment (AC2)

MS No.: essd-2023-9

MS Type: Data description paper

Title: High-resolution distribution maps of single-season rice in China from 2017 to 2022

Dear editor,

We are very grateful for your suggestions on our manuscript "High-resolution distribution maps of single-season rice in China from 2017 to 2022" (MS No.: essd-2023-9). Based on the previous revision, we further revised our manuscript according to these suggestions. Consequently, our manuscript has been improved even more.

Our detailed responses are in the supplement. Please note that the suggestions are in **bold**, followed by our responses in regular text. The revised and newly added sentences have been highlighted in red.

Sincerely,

Ruoque Shen, Wenping Yuan, on behalf of all co-authors
Email: yuanwp3@mail.sysu.edu.cn

**Reply to Editor**

**I have a few minor suggestions:**

Thanks for your suggestions. We revised the manuscript based on these suggestions.

**1. In line40, please give the full term of MODIS at the first-time appearance**

We revised the sentence and added the full term of MODIS.

**2. Since transplant is the key to identifying rice in this study, an explanation is needed before this sentence to clarify plant and transplant. For example, the days it lasts, the way it was performed (i.e. was the transplant performed manually or by machine?)**

Thank you for the suggestion, we revised and added some sentences in the method section to describe the transplanting of rice.

"The common method of rice establishment is transplanting. Rice seeds are first planted in a small field or a nursery, and then transplanted to the main field after the rice seedlings reach the three-leaf stage. The transplanting method can be divided into machine transplanting, manual transplanting, and seedling-throwing. All the transplanting methods require the field to be flooded, which is the main feature that distinguishes rice from other crops." (Lines 144–147)

**3. Besides, it has been reported that direct-seeded rice (DSR) is increasingly used. For DSR, rice seeds are sown directly into the field, as opposed to the traditional method of growing seedlings in a nursery, then transplanting them into flooded fields. If DSR was adopted, then the approach used might fail to detect rice areas. These should be discussed.**

We added several sentences in the Discussion about direct-seeded rice.

"In recent years, due to the shortage of rural labor, direct-seeded rice (DSR) has been increasingly used in China (Chakraborty et al., 2017). Unlike transplanting, DSR does not require seedling raising and transplanting. Instead, the seeds are sown directly in the main field. Depending on the field conditions, there are three types of DSR: wet direct seeding, water direct seeding and dry direct seeding (Farooq et al.,

2011). The wet direct seeding sowed the seed in puddled soil surface, and the water direct seeding sowed the seed in flooded fields. Most of the DSR belongs to these two types. In contrast, the dry direct seeding sowed the seed in a dry field. Therefore, our method can be used to identify rice fields of wet or water direct seeding by capturing the moisture or flood signal, while rice fields using dry direct seeding cannot be identified using our method. Some studies have also pointed out that certain types of DSR may have a weak flooding signal compared to transplanting, making it difficult to distinguish them from other crops using traditional classification methods (Guo et al., 2019). At present, the proportion of dry direct seeding in China is small, and it has a limited impact on the accuracy of the distribution map. However, as dry direct seeding continues to spread, its impact on rice mapping will become difficult to ignore. New methods for rice mapping must be developed in the future." (Lines 312–323)

**4. I think a conceptual flow chart will be greatly helpful for readers. Could you add a conceptual figure to describe the method used?**

We added a conceptual flow chart in the method section.

"Figure 3 shows the flow of the single-season rice mapping method proposed in this study, including four steps: (1) preprocess of the Sentinel data; (2) calculate the distances of SAR and optical bands separately using the TWDTW method with translation and stretching; (3) combine the distances of the two bands using a weighted sum; (4) generate the distribution map using a threshold determined by the provincial-level statistics." (Lines 125–128)

[Figure]

**Figure 3:** The conceptual flow chart of the method.

**5. The insert in the figures should be adjusted to cover the north of 23 N, and better show the entire Taiwan area.**

The maps in the manuscript were created following the requirements of the Standard Map Service of the Ministry of Natural Resources of the People's Republic of China (http://bzdt.ch.mnr.gov.cn/index.html). According to the standard, the northern boundary of the inserted map should be close to the Tropic of Cancer and should not include the entire island of Taiwan.

**6. It seems Figure 10 was not cited in the main text. Please check.**

We checked and found that we did not miss the citation of Figure 10, it was in Lines 230–234 of the previous revision. Since we added the flow chart as Figure 3, the Figure 11 is now cited in Lines 237–241 of this version.

**References**

Chakraborty, D., Ladha, J. K., Rana, D. S., Jat, M. L., Gathala, M. K., Yadav, S., Rao, A. N., Ramesha, M. S., and Raman, A.: A global analysis of alternative tillage and crop establishment practices for economically and environmentally efficient rice production, Sci Rep, 7, 9342, https://doi.org/10.1038/s41598-017-09742-9, 2017.

Farooq, M., Siddique, K. H. M., Rehman, H., Aziz, T., Lee, D.-J., and Wahid, A.: Rice direct seeding: Experiences, challenges and opportunities, Soil and Tillage Research, 111, 87–98, https://doi.org/10.1016/j.still.2010.10.008, 2011.

Guo, Y., Jia, X., Paull, D., and Benediktsson, J. A.: Nomination-favoured opinion pool for optical-SAR-synergistic rice mapping in face of weakened flooding signals, ISPRS Journal of Photogrammetry and Remote Sensing, 155, 187–205, https://doi.org/10.1016/j.isprsjprs.2019.07.008, 2019.